# The practice of sustainable fashion of luxury boutique fashion brands in Vietnam: What go right, and what go wrong

**Khai T. Nguyen[1], Phuong Ngoc-Duy Nguyen[2,3]\*, Long Thang Van Nguyen[4], Rajkishore Nayak[4], Thang Q. Nguyen[5]**

1 School of Business Administration, Ho Chi Minh City Open University, District 3, Ho Chi Minh City, Vietnam, 2 School of Economics Finance and Accounting, International University, Thu Duc City, Ho Chi Minh City, Vietnam, 3 Vietnam National University Ho Chi Minh City, Thu Duc City, Ho Chi Minh City, Vietnam, 4 School of Communication & Design, RMIT University, Vietnam Campus, District 7, Ho Chi Minh City, Vietnam, 5 Faculty of Tourism & Hospitality Management, HUTECH University, Ho Chi Minh City, Vietnam

\* nndphuong@hcmiu.edu.vn

**Data Availability Statement:** All relevant data are within the paper and its Supporting Information files.

**Competing interests:** The authors have declared that no competing interests exist.

## Abstract

This study investigated the current practices and challenges for the sustainable fashion of luxury boutique fashion brands (LBFBs) in Vietnam. A series of in-depth interviews with 20 founders and managers of LBFBs in Vietnam was conducted. Findings show that sustainable practices improve ethnic cultures, strengthen the usage of local resources, promote sustainable lifestyle, and thereby contributing to sustainable development of the boutique fashion brands. However, the brands face some challenges while dealing with their stakeholders such as shortage of available internal resources, bias in consumer perception and purchase behaviors, and legal barriers to achieve accredited environment certification that, in turn, weaken the sustainable practices in the local context. Results also provide some insightful information for small & medium sized enterprises (SMEs) to adjust their sustainability practices in order to improve their competitive advantages in the marketplace.

## 1. Introduction

The global luxury fashion market has witnessed tremendous growth, where the sales of high-end apparels, watches, and jewelry have substantially increased with an average growth rate of 11% during 2019 [1]. The global luxury fashion market have been projected to grow 3–5% a year through 2025, reaching $426–486 billion USD [2]. The key luxury fashion market is no longer solely existed in the Western countries, but recently expanded to new emerging markets such as China and Vietnam [3,4]. Vietnam is one of the emerging countries in the global fashion and textile supply chain [5]. Moreover, Vietnam's domestic market has remained one of the ideal markets in the global luxury fashion sector [6].

In spite of the impact of Covid-19, luxury fashion brands are increasingly entering to Vietnam targeting the young population and burgeoning middle class, and the market has not suffered much financial loss [4]. While customers know the luxury fashion through high-profile

brands such as Louis Vuitton, Hermes, Burberry and Gucci, the luxury fashion market also includes boutique fashion businesses where a team of local skillful artisans and craftsman produce luxury and customized apparels, jewelry for high-end customers [7]. In Vietnam, there has been substantial rise of many global luxury fashion brands in addition to the emergence of local boutique fashion brands or LBFBs [8].

LBFBs refer to small and medium-sized fashion enterprises (SMEs) involved with the designing, manufacturing, and selling of stylish clothing, jewellery, or other luxury goods. LBFBs represent a relatively small sector compared to the global luxury fashion brands and even fast fashion brands. Unlike the global luxury brands, LBFBs are limited in product styles, operate in smaller spaces, and deal with smaller production volumes [9]. Therefore, the product cost may be higher, which necessitates careful sourcing of inventory to manufacture the luxury products [10]. Some LBFBs manufacture their own products with authentic in-house designs using luxury materials [11].

As the number of customers for LBFBs are limited, building a strong luxury brand image is more challenging for boutique fashion brands in comparison to high-profile luxury brands (HPLBs) [12]. Although both the brands have a number of common characteristics, large HPLBs have strong brand equity, well-established products/services and distribution channels [13], which makes the marketing of luxury fashion easier. As such, LBFBs need to find effective strategies for their survival, growth, and profitability [8].

Recently, the sustainability concept has been gaining impetus in fashion business operations and marketing activities in order to differentiate the brands and have more competitive advantage in the marketplace [14]. Customers are more concerned whether the luxury fashion brands they purchase follow sustainable practices during manufacturing and supply chain processes [15]. Accordingly, many LBFBs have tried to address this concern by integrating sustainability concepts into their business strategy and operations [16,17]. However, recent studies on fashion sustainability mainly focus on the HPLBs [18,19], whereas the research on the LBFBs has been limited.

Moreover, most studies on sustainable fashion were conducted in developed countries (such as the US and UK) discussing sustainable developments in their global fashion supply chain [20–23]. Only a few studies have reported the sustainability status of fashion supply chain in emerging countries such as Vietnam, who plays an important role in the global fashion industry [24,25]. There have been a few attempts to develop the theoretical model of sustainability of small and medium businesses, however, most of them are conceptual studies [12,26], which lack empritical evidence to verify how the practices work in the emerging countries. Accordingly, an attempt has been made in this paper to investigate the sustainable practices in Vietnamese LBFBs. To meet the objectives, this research has attempted to investigate the sustainable practices among the LBFBs in Vietnam with the following two research questions (RQs):

RQ 1: *What are the current sustainable practices followed by the Vietnamese LBFBs*?

RQ 2: *What are the major challenges faced by the Vietnamese LBFBs while working with the relevant stakeholders to run their business sustainably*?

The manuscript has been organized as follows. We first review some key literature in sustainable practices and theoretical foundations for luxury fashion. We then provide details of the research methodology and discuss the research findings that answer the research questions. The paper concludes with a discussion of the research findings and highlights future research directions.

## 2. Literature review

### 2.1.Sustainability in fashion industry

With the increased emphasis on reducing the global warming and climate change, sustainability concepts have gained significant importance in the last two decades [27–29]. The term sustainability was coined in 1987 in Brundtland report, which means satisfying the current needs without compromising the future generation's needs [30]. Sustainability is viewed from three perspectives: environmental, economic, and social, which are known as the "Triple Bottom Line (TBL)" [29]. [31] highlighted that sustainability involves respecting people (at all levels of the organization), and the community; respecting the planet, recognizing that resources are finite; and generating profits that arise from adhering to these principles.

The first aspect, environmental sustainability relates to the impacts caused due to a large amount of energy usage and water consumption; greenhouse gas (GHG) emission; hazardous waste generation; and discharge of toxic effluent containing dyes, finishes and auxiliaries to the eco-system during garment manufacturing [32]. Fashion manufacturing has been recognized as one of the largest environmental polluters as several processes use large quantities of chemicals, water, and auxiliaries [33].

The second aspect, social sustainability traditionally relates to the improvement of working conditions; working hours; avoiding racism, gender equality, fair wages; health and safety of employees [27]. Sustainability goes beyond the relationship with the environment. It should address issues of the society, local communities, and relationships within our business operations. Thus, sustainable businesses must reduce the negative impacts on human livelihoods and well-being, with intersecting ecological, economic, and socio-political dimensions [34]. The pandemic has emphasised the importance of social sustainability that ensure work-life balance for employees, local community development, and community support during and after Covid-19 to ensure healthy livings of relevant stakeholders [34,35]. However, the social aspects of sustainability have also been neglected in many countries manufacturing fashion.

The third TBL or economic aspects of fashion sustainability is related to how the business operation impact on the overall economy health of its support networks and community [36]. This aspect in fashion is mostly related to using resources in a controlled manner so that the manufacturing of fashion can be sustained indefinitely. [37] highlighted that "The major challenge in fashion sourcing lies how the souring process can be ethical and transparent, in addition to the environmental pollution, and ultimately the garment's aftercare and disposal."

Due to the complex nature of the supply chain of fashion and textiles, fashion brands fall a prey in the trap of sustainability knowingly or unknowingly [23]. However, the successful application of sustainability concept in business operations and marketing activities could help fashion brands to differentiate its product/services and have more competitive advantages in the marketplace [38]. Many global fashion brands are switching to sustainable products and processes, which is helping them to market their products [39,40]. In order to be sustainable, all fashion brands need to conserve raw materials, adopt safe manufacturing routes to avoid environmental pollution, while utilizing corporate social responsibility activities [41].

In particular, for luxury fashion market, all brands have been under pressure to tackle the sustainability consciousness [19]. Luxury brands including fashion brands have been defined as 'having high quality, offering authentic value, conveying a prestigious image, being worthy of a premium price, and inspiring a deep connection' [42, p,571]. While it is a small sector in fashion industry, luxury fashion is considered as the most profitable and fastest-growing brand segment [43]. The development of economics provide more opportunities for luxury fashion to grow faster with two or three digits annually [44]. Customers buy luxury fashion brand for exclusivity, status, quality, and high price [45]. The usages of luxury brand can help them to express their social identify following the success in their career and life [19].

Luxury fashion and sustainability or ethical practices are contradictory based on the values of altruism, as luxury fashion indicates wastefulness, ostentation, resource oriented, superficiality and thoughtlessness [45]. In the past, luxury fashion was indicative of developed economy, however, the recent luxury fashion market has spread to many of the developing countries such as China, India and Vietnam [46]. Out of these emerging economies, Vietnamese luxury fashion segment is of special significance as Vietnam is the leading exporter of fashion to around the world [27] such as the US, EU, Japan, and South Korea. The consumers of luxury fashion are increasingly emphasizing on the environmental and social impacts of the luxury goods they purchase [47]; hence, the luxury fashion brands are integrating sustainability and ethical practices for their brand management [48].

Since the public's concerns about sustainability matters, all HPLBs are under tremendous pressure to be more transparent of the business operations. To respond to the increasing interests, luxury fashion brands have engaged in a continuous process to transform the value chains sustainably from sourcing of raw materials, logistics and delivery, production process, retailing, and finally, post-purchase recycling with 3R concept (Reduce, Reuse, Recycle) [45,49]. Some luxury groups (such as Kering) show the transparency by publishing annual sustainability reports, alongside their Annual Financial Reports [50]. Moreover, many major jewelry brands agree to sign Kimberly agreement to commit not selling gems coming from war zones (Burma, Zaire, Congo, etc.) [51]. Some practices of sustainable concept were reported at number of large high-profile luxury fashion brands such as The Body Shop [52], Marks and Spencer [38], and H&M [40]. While scholars pay more attention for the practice of sustainable fashions of large high-profile fashion brands (see Table 1 for summary), there are limited attempts to explore the sustainability concept among LBFBs.

LBFBs represent a relatively small sector compared to the HPLBs. Similar to the HPLBs; waste generation, excessive usage of natural resources, and negligence of the workforce are some of the major challenges that LBFBs needed to deal with [53]. Many LBFBs are reluctant to incorporate sustainability concepts into their business model due to consumers' demand for good quality and stylish clothing at cheaper price [33]. Therefore, LBFBs were claimed for not caring of the TBL of sustainability that has negatively impacted to their stakeholders [36].

Vietnam houses about 6000 textile and garment industries, large proportions of which are SMEs. The SMEs in fashion including LBFBs are categorised into the privately-owned industries, which are decentralised and smaller in size with fast-decision-making process. They cover more than two third of fashion jobs and become crucial contributors to poverty alleviation in Vietnam [54,55]. Fashion products from LBFBs are stylish, unique, together with personalized customer services with an intimate authentic cultural experiences [36]. The more popular is the LBFB in the marketplace, the more vulnerable is the brand towards public criticism over their operation. Since many of the HPLBs have adopted new technologies, using sustainable products and processes; and taking care of their employees in the drive of sustainability [33], it is essential for the LBFBs to follow sustainable models by simultaneously delivering products manufactured with a proper consideration of the TBL of sustainability.

## 2.2 Sustainable fashion for LBFBs and resource dependence theory

The resource dependence theory [56] were used to explain the sustainability practices of LBFBs. The resource dependence theory highlights that resources are defined as 'anything that could be thought of as a strength or weakness of a given firm' [56, p.172], which is the basis of effective business operation and stakeholder dependences. The theory proposed that the business cannot operate well without critical resources such as labor, capital, policies, technology, and raw materials [57]. These resources are managed by other stakeholders, identified as any

**Table 1. Key studies of luxury fashion and sustainable concept.**

| References | Organization type | Areas of study | Findings |
|---|---|---|---|
| Achabou and Dekhili [45] | French fashion brands | Investigates the degree to which sustainable developments can be associated with luxury products. | • The use of recycled materials in luxury products negatively affects consumer preferences of the brand, which creates a certain incompatibility in recycling of luxury products.<br>• Consumers of luxury fashion brands primarily consider the intrinsic quality of the product although they are concerned about the impact of the product on planet, which is the secondary selection criteria. |
| Joy, Sherry [15] | Large fashion brands | Discusses luxury fashion brand's approach on quality and environmental sustainability | • Luxury brands can become the leaders in sustainability because of their emphasis on artisanal quality.<br>• Luxury brands have high ethical standards in sourcing, efficient use of material, low-impact manufacturing, assembly, and distribution. |
| Kapferer and Michaut-Denizeau [72] | Large luxury fashion brands | Explains why customers are not concerned about sustainability considerations when they purchase a luxury product | • Sustainability has become an element of quality expected by luxury customers.<br>• The buyers of luxury fashion have ambivalent attitudes, such that they consider luxury and sustainability somewhat contradictory, especially about the social and economic harmony facet of sustainable development.<br>• There is a high risk for brands that ignore the sustainability requirements as the brands can be vulnerable to criticism because of their visibility. |
| Han, Seo [73] | Large fashion brands | Explains how luxury brands can encourage consumers to be oriented towards sustainable fashion consumption | • Developing and staging memorable consumer-centered experiences helps to orient consumers toward sustainable fashion.<br>• Traditional communication tools (i.e. advertising and public relations), may not foster sustainable fashion consumer literacy. |
| Nagurney and Yu [74] | Large luxury fashion brands | A model was developed for fashion supply chain under oligopolistic competition including differentiated products and environmental concerns. It was assumed that each fashion brand has unique product and there is a competition among firms until an equilibrium is achieved. | • Each fashion firm seeks to maximize its profits as well as to minimize its emissions throughout its supply chain with the latter criterion being weighted in an individual manner by each firm. The competitive supply chain model is network-based and variational inequality theory is utilized for the formulation of the governing Nash equilibrium as well as for the solution of the case study examples |

group or individual who can affect or is affected by the business operations [58]. Since some stakeholders (e.g., the suppliers, consumers, competitors, the authorities) control certain resources vital to the business operations, the business need resources from other stakeholders in order to execute any business plan. Conversely, the other stakeholders also need resources from the business to satisfy their interests or address their social, economic, and political issues. This results in interdependency, where each stakeholder depends on others and has influence over the others [59]. In order to generate appropriate resources, the business must be involved in the negotiation process with other stakeholders on the win-win situations. This help the business to minimize the risk of its execution plan, and reduce uncertainty in the marketplace, while also maintaining stability in operations [60].

The merits of resource dependence theory can be extended to the sustainability in luxury fashion in explaining the interdependence between the LBFBs and their relevant stakeholders. To ensure the sustainable model for business development, the boutique fashion business needs to focus on more stakeholder participations and collaborations to handle the interests in social, economic, and environmental issues [41]. On one hand, the stakeholders rely on the business to obtain updated information about the marketplace and seek for sustainable practices in place. On the other hand, the business relies on its members' contributions for its survival and sustainable development. The provision of stakeholder resources such as shared

experiences, brand knowledge and other expertise can enhance the productivity of sustainable practice. This, in turn, increases the legitimacy of the sustainable activities, especially in terms of addressing stakeholder concerns about sustainability. But this process of bargaining, negotiation, education strategies among all stakeholders bring about symbiotic changes in the ideas, attitudes, and behaviours of both the organization and its publics [61]. As a result, resource contributions from consumers can strengthen the overall performance of sustainable business model.

## 3. Methodology

The main aim of this research was to examine the current practices and challenges, which LBFBs encounter in exercising the sustainability concepts for gaining competitive advantages against the HPLBs. Such a study requires careful selection of information-rich cases with valid, reliable and practice focused data. As such, the study employed the interpretive research approach with the use of case study method to gain more insight, accuracy, and depth on the current specific research issues of sustainability. Prior to conducting the survey, the University's Ethics Committee approval was obtained. In selecting the cases, purposeful sampling technique with snowball strategy was used to obtain in-depth information from the practitioners who are not only knowledgeable, but also willing to participate and voice their opinions regarding the topic being investigated.

Based on the sampling theory, ten LBFBs in Vietnam were chosen. The fashion market in Vietnam has been in the rise for the last few years, which witnessed the rise in significant number of SMEs starting their businesses. The selections of these LBFBs are based on the matching of their product lines and prices in comparison with the large HPLBs (e.g., Louis Vuitton, Gucci, and Burberry) in the marketplace. Further, their businesses products and services address the authentic Vietnamese fashion culture of handicraft and embroidery, which is alighted with the concept of sustainability.

Out of the ten cases, two (SME1 and SME2) were invited first and agreed to take part in the study. Using snowball sampling technique, the remaining eight cases were suggested by these two SMEs and agreed to join the research project. With the exception of SME1 and SME10 where only the founder was interviewed, interviews were conducted with business owners and managers of other cases. Participants were required to read and agree with the informed consent before starting the interviews. There were 20 interviews with respondents having experience in the fashion business from four to ten years. Given the research questions, the researchers decided to cease the data collection with these 20 interviews as there was no new information or new themes observed in the data (Guest, Bunce & Johnson 2006). This indicated that the data was saturated and ready for analysis. Table 2 depicts the details of participants taking part in the study.

Prior to the data collection stage, interview protocols were developed from relevant literature consisting of ten main questions. Each question was supported by several prompt questions to assist the interviewers to seek further information. Consequently, data was predominantly collected via 45–60 minutes semi-structured interviews with open ended questions to help the researchers gain information on the experience, perception, and opinions of the research participants regarding their practice of sustainability related business activities [62]. In addition, this data collection method enabled both respondents and researcher to follow up particular issues, dismiss them as insignificant, or suggest additional insights which were not sufficiently planned during the preparation of the interview protocols. Once the data was collected, it was transcribed by the researcher into MS Word documents. Since all interviews were conducted in English, no translation was required. These documents were then

**Table 2. Details of interviewees selected for case study.**

| No | SMEs Code/ Business type | Participant | Position | Experiences (years) |
|----|--------------------------|-------------|----------|---------------------|
| 1 | SME 1 / Bespoke tailoring | SME1.F | Founder | 7 |
| 2 | SME 2 / Traditional Clothes | SME2.F1 | Co-Founder | 6 |
| 3 | | SME2.F2 | Co- Founder | 4 |
| 4 | SME3 / Bespoke tailoring | SME3.F | Founder | 9 |
| 5 | | SME3.BM | Brand Manager | 8 |
| 6 | SME4 / Artisan Handicraft | SME4.BM | Brand Manager | 5 |
| 7 | | SME4.PM | Merchanting Planning Manager | 4 |
| 8 | SME5 / Traditional Clothes | SME5.F | Founder | 10 |
| 9 | | SME5.BM | Brand Manager | 5 |
| 10 | | SME5.OM | Operation Manager | 5 |
| 11 | SME 6 / Artisan Handicraft | SME6.F | Founder | 8 |
| 12 | | SME6.MM | Marketing Manager | 7 |
| 13 | | SME6.PM | Sourcing & Procurement Manager | 8 |
| 14 | SME 7 / Artisan Handicraft | SME7.BM | Brand Manager | 8 |
| 15 | | SME7.CD | Creative Director | 10 |
| 16 | SME 8 / Traditional Clothes | SME8.F | Founder | 8 |
| 17 | | SME8.MM | Marketing Manager | 5 |
| 18 | SME 9 / Made-to-measure | SME9.F | Founder | 5 |
| 19 | | SME9.OM | Operation Manager | 5 |
| 20 | SME 10 / Made-to-measure | SME10.F | Founder / Managing Director | 5 |

formatted into appropriate headings before being importing into NVivo software for the process of coding.

In analyzing the interview transcripts, it is necessary to decide on unit of analysis as it forms the boundary of the case. The unit of analysis could be either individuals, groups or organizations. As the purpose of the study was not comparing findings across the cases, the unit of analysis in analyzing the interview data was therefore the individual. Prior to analyzing data, a set of proposed categories/themes was developed from the research questions. Thematic analysis with theme-based approach was used to analyze the data by reading transcripts, identifying themes then coding the data. We followed two coding cycles approach as mentioned in the literature [63]. The first holistic coding focuses on the essence of text excerpts, while the second descriptive coding summarizes the excerpts to form subcategories of the phenomenon. To minimize the bias for coding, two researchers coded the data separately and compared the findings. Most of the coded themes were matched. New themes were discussed and further developed during this process by the research team. The final coding frame was then analyzed across interviews.

## 4. Interview findings

### 4.1 The current practices with relevant stakeholders of sustainable fashion

Majority of the respondents reported four primary stakeholders are the most important while sustainability is considered: customers (75%), suppliers (71%); local communities (61%), and employees (55%). Interviewees reported: *"Customers are aggressive about sustainable materials [. . .]. They can call our customer care to know more about the sustainability practices while manufacturing some products"*. Others reported the roles of suppliers in developing the

sustainable products that "*In the past we were 30% sustainable, this year it is 90% sustainable, and we are trying to reach 100% sustainable products in the next two years*". Some highlighted the importance of local communities: "*I'm confident that the majority of people understand about it. When you look at certain portions in community, they're more active in environmental issues*".

A founder A2 emphasized the involvements of employees in sustainable practices: "*When we engage with workers, people actually become more efficient, they become more proactive, they become more productive, they deliver more sustainable practices within their roles*". Other stakeholders reported the role of business partners, media and government agencies. But they are considered as less important due to the limited mention by the respondents (less than 50%). Through collaborations and interactions with these major stakeholders, the LBFB stabilizes their sustainable business operation by exchanging constraint resources such as materials, information, know-how, and localized customs. The inter-dependences allow LBFBs to apply sustainable practices while enabling other stakeholders to address their interests or concerns for their business survival or social welfares. Applying the iterative thematic coding process to the interview transcripts as to how participating LBFB integrate the sustainable factors into their business operations with the target stakeholders (customers, local communities, local communities, and business partners), three dominant themes were identified. The key themes include: (i) promotion of ethnic cultures, (ii) enhanced usage of local community resources, and (iii) co-creation of sustainable lifestyle.

**4.1.1 Promotion of ethnic cultures of Vietnamese minority groups.** Ethnic culture is believed to be one of the crucial influences in fashion patterns and trends. The stakeholder of Vietnamese ethnic minorities has a strong cultural background which is reflected through their traditional elegant clothes, expressive garments, costumes, and other accessories. However, they have some difficulties maintaining the current practices to represent their culture as well as improvements in their standards of living. In order to manage their uncertainty, they collaborate with LBFBs who have resource constraints for idea, production concepts for authentic clothes, garments, costume and other accessories. As such, LBFBs preserve and promote the cultural traditions of ethnic minority groups to the public through fashion product designs. LBFBs invite artisans of constraining Vietnamese minority groups onto their business production to gain their support, to use their expertise, to get legitimacy of the authentic apparel productions. In return, artisans, representatives of minority groups, ensure their cultural reflections via the design and productions of clothes, garments, costumes and other accessories.

Extract 1 (SME7): "*Fashion is taking out inspiration from the place that you're in, your environment and then to showcase it, to tell a story, to send a message. Our collection is a journey in Vietnam. We want our customers to take a piece of their travel back home.*

Extract 2 (SME5): "*We got inspiration from traditional beauty of Vietnam, the pattern that we have inspired from the shape of rice field and the unique culture of Vietnamese people.*"

Extract 1 indicates the impact of ethnic culture on fashion design at the philosophical level which could in turn be used as the general guidelines for stakeholders in creating pieces of fashion designs with unique cultural patterns. Extract 2 illustrates an explicit example as to how a particular aspect of Vietnamese culture is embedded in creating a unique fashion collection. Therefore, ethnic culture could play a key role in assisting LBFBs in differentiating themselves from competitors with exceptional products. From another angle of view, these cultural authentic pieces of design could also be seen as a means in promoting ethnic cultures to the wider world.

**4.1.2 Enhanced usages of resources from local communities.** Sustainable fashion supply chain favors the use of eco-materials in green manufacturing, distribution, and retailing. In

this case, globalization allows multinational corporations to locate the resources for their fashion production anywhere in the world to maximize economic of scales. Thus, they may have the advantages to highlight ecofriendly products for competitive advantage. With LBFBs adapting sustainability practices, the increase competitions enforce the dependence among LBFBs and local suppliers to highlight the usages of local resources in sustainable fashion productions. Local resources are sourced and used in the process of manufacturing such as using talented artisans, using local silk, fabrics, and traditional handicrafts. The use of local raw materials might help to create competitive advantage for these manufacturers via creating unique fashion products. Further, this builds up the reputation for Vietnamese ethnic ecofriendly products to attract attention from potential foreign investors and visitors to the ethnic minority travel destinations.

*Extract 3 (SME4)*: "*The limited collections made by (local) designers for instances and*, *then*, *offer the local artisans the handmade craft*. *So that I can either support designers or the artisans in those villages*"

*Extract 4 (SME6)*: "*We create everything from A to Z, from the construction of the fabric, the printing, the dyeing, the manufacturing, you know all the production unit.*"

*Extract 5 (SME8)*: "*We distribute our products in our own network of stores.*"

The above three extracts mirror how LBFBs utilize local resources in every aspect of their fashion supply chain. While Extract 3 exemplifies the making use of local skillful human resources in creating traditional authentic designs, Extract 4 and 5 showcase the roles local labor forces with raw materials play in the whole supply chain. It signifies the contribution of sustainable oriented fashion boutique SMEs to the enhancement of local community resources usage at different levels. For local communities and businesses, most are small and micro enterprises who have limited opportunities to work with the global firms in their supply chains. Instead, forming alliances with LBFBs help them to maintain the business operations and, as a result, create stable local community living environments. This practice results in a win-win situation for all involved stakeholders and, thus, contribute to the sustainable development for the brand in the local context.

**4.1.3 Co-creation of sustainable lifestyle with target customers and the brand.** The LBFBs' intention to foster sustainable fashion depends on customers because they are the predominant buyer of LBFBs. At the same time, customers are dependent on LBFBs to provide high quality ecofriendly products for their clothes selection and fashion style practices. The dependencies require both parties to co-create the sustainable practices in fashion marketplace. As such, boutique fashion businesses try to produce high quality products that can reduce the 'wear and tear' in order to decrease number of new purchases. Furthermore, these enterprises encourage their consumers to reuse and recycle their clothes by organizing swap events or reward consumers with vouchers for the next purchase. This helps to raise awareness of sustainable consumption.

*Extract 6 (SME5)*: "*We choose to convey the message of a green lifestyle to customers in various tactful and indirect ways such as clothing exchange reward programs*, *usage of the fabric shopping bags*, *and decoration of our retails stores following the ecofriendly concept*."

*Extract 7 (SME3)*: Our customers re-organize their closets to exchange them among friends and like-minded customers, or to donate for charity during the exchange event.

In addition, there are operational changes in the structure and management of the business to ensure it is sustainable. All LBFBs emphasized the importance of the integration of sustainability concept in their business operations from their supply chain (ecofriendly materials, and sustainable manufacturing) down to the retail experience (green distribution, and green retailing) to ensure the integration of sustainable concept across all aspects of the business. In the retail outlet, managers mentioned how they engage proactively in conservation efforts to

minimize the usage of water, and energy. This requires their efforts to change existing water, waste, and lighting systems. Extract 8 displays a specific example which was to remove any plastic related items in their daily activities.

Extract 8 (SME2): *"We have the recycled shopping bags, we are removing everything that is plastic such as containers, delivery, top for the cups, all the plastic bags we're using so we use recycled bags or paper bags."*

With the LBFBs, sustainability practice is moved to further levels where not only the use of fabric shopping bags is encouraged, but also the motivation of these practices to relevant stakeholders are also taken into consideration. While the activities aim to generate little or no environmental impact, they may generate higher costs on the businesses to date and create some concerns from the business partners and employees. The dependences of these internal stakeholders require the owners/managers of LBFB to stabilize the business operation with through socializing, discussing regarding valuable benefits for the brand, society, and environment.

Extract 9 (SME9): *"Because sustainable is a trend, to follow the trend you must have a team to really understand the sustainable concept, you need equipment, you need lots of resources, assets to be sustainable which require some supports from business partners and employees."*

Extract 10 (SME4): *"The production of using 0% chemicals, we need to really drive to ensure that we have got the best product to our consumers but also do no harm in the process for the workers."*

Further, the LBFBs need to convince suppliers since they do not see the installation costs of some equipment for eco-production as feasible. As such, the LBFBs' team must find the suppliers meeting the ecofriendly requirement. Extract 11 exhibits the basic practice of sustainability adaption exercised by the cases participating in the current study.

Extract 11 (SME6): *"Sourcing team and Design team have to find out the materials that is friendly to the environment."*

## 4.2 Challenges of sustainable fashion while dealing with relevant stakeholders

Another aim of the research was to examine what are the challenges faced by participating LBFBs in adapting sustainability practices. In analysing the interview transcripts, three dominant themes regarding the challenges were identified, namely: (i) Small and Medium-sized Enterprise (SMEs) related challenges, (ii) consumers related challenges and (iii) macro levels related challenges. While particular resources from these stakeholders may only constitute a small part of total resource needs, they threaten the ability of LBFBs to continue sustainable fashion practicing in the absence of the resources.

**4.2.1 SMEs challenges with internal resources.** LBFBs are considered as SMEs. They are characterised by several unique factors including the scarcity of resources and simple organisational structure with less standardised policies and procedures. For SMEs to be able to compete with the HPLBs in the fashion industry, they would, therefore, focus on the quality of their products with unique designs and service in their target market. Difficulties faced by some fashion SMEs are, hence, amplified by the lack of both talent and financial resources. The quality and unique of their products are either dependent on few key employees, owners/managers or rely on the suppliers from third parties as explained in Extract 12.

Extract 12 (SME3): *"We don't have a team specializing in recycling. We don't have a complete circle of recycling things. It prevents us from doing campaigns about exchange or recycled clothes."*

In addition, with less professionalism in working policies and procedures, it is questionable if SMEs can adapt to the changes in sustainable trends in the fashion industry and to keep

their product quality consistent. Due to the resource uncertainty, fashion SMEs in Vietnam might expose to risks of having lack of trusts from customers. The following comments extracted from the interviews could be used to confirm the above observations. Although other types of limited resources such as sustainable raw materials or finance are mentioned, the shortage of human resources with skills and knowledge regarding the sustainable practice are proved to be the central issues for most of the cases in the study.

Extract 13 (SME1): *"All the brands themselves have great strategy to communicate about sustainability. But the team in Vietnam and the market is not ready. So, if you keep saying something they don't understand, they will get annoyed."*

Extract 14 (SME6): *"The sustainability procedures are not followed strictly; many organizations take the advantage. We can't monitor 100% the manufacturing process."*

These challenges are summarised as: (1) Lack of skilful staff members; (2) Lack of financial resources, (3) Lack of professionalism for sustainable practices and 4) Limited and unstable supply for sustainable materials. The observations from collected data are consistent with previous finding from [64].

**4.2.2 Consumers' related challenges.**    The sustainability has increasingly become a standard practice worldwide. It is, however, a complicated and relatively new concept in Vietnam. Vietnamese customers are, therefore, still late adopters resulting from having insufficient knowledge about sustainable fashion. The lack of sustainable fashion knowledge limits their selections in making fashion purchase decisions, they focus on the 'beauty' of the products rather than the 'ecofriendly features' integrated in these items. Furthermore, they are mostly in favour of seeking for fashionable items which are trendy, stylish, well-fitting, and comfortable. In addition, young Vietnamese customers tend to be more with low priced items. On the contrary, as previously reviewed, fashion designs from Vietnamese SMEs following sustainability trends and guidelines often present themselves with limited styles and at high prices. The Vietnamese consumers, therefore, are lacking motivation in purchasing sustainable fashion products.

Comments from the respondents are consistent with the above review. For instance, Extract 14 highlights that customers in fact lay their prioritised focus on aspects but not the unique cultural design of the products following the sustainability practice. Further, sustainability is seen yet to be a decisive factor in making purchase decisions as being pointed out in Extract 15.

Extract 15 (SME9): *"They don't buy because it's ethical, but they buy it because it's beautiful".*

Extract 16 (SME10): *"In Vietnam, when you talk too much about sustainability, you become a hypocrite. People do not really get the concept and have the concept. It is not in their mind yet."*

Extract 17 (SME1): *"In fashion, the trends are changing, rapidly, so the products are easily out of date. It might not be trending anymore. So, it is also a problem in fashion."*

Extract 18 (SME3): *"The challenge now is here you make a lot of effort to make it organic, but people don't understand the differences. They don't get it."*

The findings are consistent to [65] who found that customers may care more about the price and appearance of clothing than its sustainable benefits of whether they are made by organic cotton, or the low carbon production process has been used. The customer related challenges are summarized as: (1) Lack of knowledge about sustainable fashion, (2) Lack of motivation such as limited styles and less stylish and (3) Lack of financial resources.

**4.2.3 Macro level related challenges.**    Analysing the interview transcripts revealed factors which are from the macro levels. There exist legislations, regulations, and guidelines from the Vietnamese government to assist organisations to obtain required certifications. These certifications are used to indicate the trustworthiness of the products being claimed to follow the sustainability practice in manufacturing. However, fashion SMEs face challenge as the

certification processes are either too complicated or time-consuming or over the budgets of SMEs. In fact, the guidelines are most suitable for larger fashion businesses but not for SMEs, who might need assistance from the government. Further, the presence of sustainability education or activities is also mostly absent from the curriculum in moving the awareness to the level beyond being just 'environmental protection'. Extract 18 and 19 from the interviews clearly support the above observation.

Extract 18: *"Vietnam do have to improve its social standard according to the EU (European Union). However, I'm afraid that these standards are too high compared to the TPP (Trans-Pacific Partnership) one and how can we raise the awareness of stakeholders and increase the productivity of the employees."*

Extract 19: *"We're very sustainable but the certification only applies in mass companies. How to communicate on sustainable fashion. It is difficult for me. I've been talking to different agencies, and I can't find more information. I don't know how to do it in a subtle way."*

From the above analysis, two challenges at the macro level include: (1) Lack of supporting infrastructure for SMEs and (2) Low level of sustainability related education activities. These challenges are beyond the scope of the LBFBs to handle, it shows the dependences on government supports for policy implications and practical sustainable education activities.

## 5. Discussion

Fashion industries are behind other industries in applying sustainability concept [65]. To address the criticism towards sustainability, luxury fashion businesses including LBFB must focus on ecofriendly practices in their supply chain together with generating resources for social and economic development. Our aim in this study was to examine the current practices of sustainability among LBFBs and some challenges that the business has to deal with. Our findings show that LBFB integrated the concept of sustainability in their business models and found the practices to be useful for the healthy brand development. Three major themes from the interview data were discussed: promotion of ethnic minority culture, extensive usage of local resources, and the co-creation of sustainable lifestyle.

The findings indicated that the sustainable fashion practices have been adopted within the participating cases throughout most of the activities from product design to supply chain related activities. In fact, as implicitly indicated in Extract 1 and 2, the sustainability is to start at the product concept level in which if it is guided by cultural values and beliefs, eco-friendly fashion products will be foreseeable in return. Further, via the use of local raw materials and resources such as local silks with ethnic artistic talents, the finished goods are not only beneficial from their unique features but also deemed to meet environment and social requirements for being sustainable (Extract 3 and 4). In these cases, the sustainability advancements are also integrated in their daily activities (Extract 7, 8 and 9) such as eliminating the use of plastic bags.

The modern life changes the way how publics understand and maintain the ethnic culture. A number of ethnics' communities struggle to maintain their cultural background, while trying to keep up with the development of the economy. They expect the public to be aware of the beauty of their culture, and thus, they need some supports from the business to change the way their culture is presented. However, the support is rarely seen from the high-profile fashion brands which are claimed to promote the Western luxury lifestyles. The LBFBs personalized their identity into local ethnic culture. In the case of Vietnam, the design and materials were now inspired from the local ethnic communities, particular in the Northern Vietnam. By collaborating with the local artisans to create unique clothing lines, the luxury boutique brands try to blend and upscale the traditional indigenous clothing styles with modern design

concepts in order to promote the diversity of Vietnamese ethnic culture. This fusion of modern and traditional ethic styles has led to an outburst of interest in cultures among target customers, which were once ignored or underdeveloped by luxury fashion brands around the world.

Moreover, attracting more customers to buy the brand products could contribute to the local community's social and economic revitalization and economic diversification. In turn, job creation and additional training are the offers that LBFB can contribute to the local ethnic communities. The brands offer artisans opportunities to maintain their traditional works, increase the training and recruitments of local people to work for the LBFBs, which help them to maintain their social and traditional cultural practice while earning reasonable incomes to support their family and children's education. These supports have been extended to conservation and restoration of the traditional village and community. As such, the LBFBs how their businesses have been grown sustainably together with the ethnic communities. Thus, our findings enrich the works of [66] and [67] who suggest the fashion brands should be openness into different culture, stimulate appreciation of other humanities, and integrate the concept in their business operation.

In addition, with the limited resources of the planet, more publics aim to enhance their practice of reducing their demands. As such, the LBFBs' aim to co-create sustainable lifestyles by adopting the concept of 'reduce, recycle, reuse (3R)'. Customers are encouraged to recycle their clothes by returning their clothes to the store and sharing with the people in need. However, differed from the big brands' recycling practice [40], LBFBs extended this recycling practices to exchange activities among their customers. Customers feel free to share their experiences about sustainable lifestyle to other like-minded people, who lack sufficient information about how to do this. Further, the event allows them to swap their clothes, to recycle their clothes and to donate their clothes. In order to convince the customer for the sustainable lifestyle, LBFBs have tried to integrate the sustainability concept into their business and supply chains such as using costly but ecofriendly and recycled materials, natural lightings, saving water, plastic-free packaging, and creating less wastes. Their businesses no longer compromise with harmful resources for making profit. The findings enrich the works of [68] and Lee and DeLong [69], who reported that businesses should set the standard for customers and promote sustainable consumption, and encourage consumers to avoid impulse buying through experience sharing, networking with others which, in turn, foster the sustainable habits among them.

While sustainable practices have been applied successfully in the LBFBs, there are some challenges that limited the enhancement of the practices in some businesses. The results suggest that, despite the existent differences among LBFBs (full controller of supply chain, or no controller of supply chain), some challenges were discussed and agreed among them namely limited skillful staff members and limited financial resources, lack of professionalism for sustainable practices and limited and unstable supply for sustainable materials. Other concerns are lack of customer understanding, negative perception among other stakeholders, and lack of supports from the authorities.

The marketing theory of double jeopardy argue that the small brands have lots of disadvantage over the big brands. The HPLBs have more customers, larger market shares than the LBFBs, which in turn, generate higher trust level among stakeholders for the ethical practices for their products [70]. This is illustrated through our findings from LBFBs, when their small brands struggled to survive due to lack of supports and understanding from stakeholders. First, while HPLBs have strong logistical and supply chains worldwide to utilize enough resources to support for their sustainable productions and operations, LBFBs aim to develop the local supply chain to support small producers and their communities. However, a few issues are discussed including higher prices, unavailable ecofriendly materials for productions

in the local area. Due to their business size, it is hard for them to convince the large, certified suppliers to provide sufficient ecofriendly materials for their productions within their budgets and timeline. In summary, these challenges reduce the efforts of LBFBs to incorporate sustainability concept in their business successfully.

## 6. Theoretical contributions and practical implications

Although previous research has identified the applications of resource dependence theory, quantitative survey is the most common method used to collect data and test the theory via conceptual frameworks. There are limited empirical work that have looked at this theory in the qualitative lens to provide richer information for interdependences among stakeholders. Building on recommendations from [71], this study made an important contribution to the literature by offering a resource dependence perspective to understand how interdependences works with SMEs in Vietnam's luxury fashion sector. Due to the small scale of business, LBFBs have some limitations for business developments in order to compete with the global fashion brands. Further, their sustainable development requires lots of resources (e.g., financial, technical, and manpower) that they may lack. Thus, they will be very likely to undertake new ways to solve problems via networking and collaboration.

Resource Dependency Theory argues that the fewer the number of resources, the more numerous the connections and interdependencies among stakeholders. Further, there are two angles of interdependences: 1) filing the resource gaps of the business, and 2) improving the usages of existing resources. In this study, collaborations with relevant stakeholders will help the brand to fill the gaps of resource constraints. New knowledge, expertise and resources from local community businesses, artisans, employees enable LBFBs to develop their ecofriendly, authentic fashion products. Moreover, LBFBs work well with their customers to cocreate the sustainable lifestyle via good product quality, clothes exchange programs, and ecofriendly store concept integration. The interdependencies will help LBFBs to overcome the disadvantages of small size and limited resources to better address the needs of customers. On the other hand, working with LBFBs allow relevant stakeholders to overcome obstacles associated with their limited network and market knowledge. It helps local businesses, Vietnamese minority groups to improve the usage of their existing resources (labour forces, authentic designs, local silk, fabrics, and traditional handicrafts).

Regarding the challenges, the dynamic nature of external market environment requires LBFBs to seek more supports from other stakeholders. Thus, holding some valuable resources was a necessary but insufficient condition for the business to achieve competitive advantages in the marketplace [71]. To address these issues, LBFBs need additional efforts to educate customers for the practice, to seek helps from the government for the certification and supporting policies to convert resource portfolio into their capabilities in the marketplace. In the meantime, they still struggle to maintain their competitive advantages in the marketplace while doing good enough for the sustainable development. These results highlight the implication of Resource Dependences Theory in the context of sustainable fashion SMEs.

There are few practical implications from the study. First, the development of sustainable fashions are challenging for LBFBs due to intense competitions from the HPLBs. The LBFB businesses are often unable to obtain sufficient resources for sustainable business operation. Instead, LBFBs should possess resources from relevant stakeholders including local businesses, and artisans from minority groups (manpower, authentic designs and fabrics with traditional handicrafts). LBFBs can benefit from these stakeholders' participation in the supply chain for the product design and quality improvements. LBFBs should take advantage of business

partner knowledge to improve their business operation and capabilities and better align products with local market trends.

Second, the improvement of product quality and customer services could increase customer engagement with the brand to co-create the sustainable lifestyles. Durable products not only help customers reduce their frequent purchase, but it also encourages them to join clothes exchange programs organized by LBFBs, to share the sustainable concept with other like-minded people and, in turn, can promote the brand implicitly. It allows customers to reduce their fast fashion purchase habits and be more sustainable for fashion product consumptions. Third, while there are some good sights for the sustainable fashion outlooks for LBFBs in Vietnam, they should know the existing challenges such as constraint of ecofriendly materials in the local community, sustainable fashion practice, readiness among mass customers, and limited supports from the authorities. If LBFBs want to scale up their business, they can't simply rely on local resources for productions and operation. Many local businesses are not ready with the larger volume for productions with consistent good quality. Moreover, they should spend more efforts to educate customers for sustainable fashion concept as well as spend more resources for some certificates. Thus, maintaining the current scales of business with niches customer segments maybe a good option for LBFBs to grow slowly but sustainably in this market condition.

## 7. Research limitations and future research

Some limitations for the paper are discussed in this section. First, the study only examined the practices at boutique fashion brands in Vietnam which may reflect the cultural context of the country. Further research from different emerging countries could extend our understanding of the sustainability in LBFBs. Second, most of the businesses that we interviewed belong to apparel categories, which may not represent the full picture of luxury fashion. Further research should be conducted into other luxury categories such as watches, furniture, winery, or other luxury goods, and services. Third, the study examined sustainability concept form the brand perspective only. The sustainable concept may be perceived differently from other stakeholders such as customers, authorities, media, and fashion suppliers. Also, it is beneficial to extend the interviews to consumers to have a complete picture regarding the customer-related challenges. It would be interesting to further conduct quantitative studies with these stakeholders to widen our knowledge about the sustainability of luxury brands.

## Author Contributions

**Conceptualization:** Khai T. Nguyen, Long Thang Van Nguyen.

**Investigation:** Long Thang Van Nguyen.

**Methodology:** Rajkishore Nayak.

**Project administration:** Phuong Ngoc-Duy Nguyen.

**Validation:** Thang Q. Nguyen.

**Writing – original draft:** Long Thang Van Nguyen.

**Writing – review & editing:** Phuong Ngoc-Duy Nguyen, Rajkishore Nayak.

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
