## [Decision Letter · Decision Letter 0]

21 Mar 2023

PONE-D-23-01713The Practice of Sustainable Fashion of Luxury Boutique Fashion Brands in Vietnam: What go right, and what go wrongPLOS ONE

Dear Dr. Nguyen,

Thank you for submitting your manuscript to PLOS ONE. After careful consideration, we feel that it has merit but does not fully meet PLOS ONE’s publication criteria as it currently stands. Therefore, we invite you to submit a revised version of the manuscript that addresses the points raised during the review process.

We look forward to receiving your revised manuscript.

Kind regards,

Han Lin

Academic Editor

PLOS ONE

3. Please provide additional details regarding ethical approval in the body of your manuscript. In the Methods section, please ensure that you have specified the name of the IRB/ethics committee that approved your study.

4. PLOS ONE does not copy edit accepted manuscripts (https://journals.plos.org/plosone/s/criteria-for-publication#loc-5). To that effect, please ensure that your submission is free of typos and grammatical errors, including in the title.

“NO”

7. In your Data Availability statement, you have not specified where the minimal data set underlying the results described in your manuscript can be found. PLOS defines a study's minimal data set as the underlying data used to reach the conclusions drawn in the manuscript and any additional data required to replicate the reported study findings in their entirety. All PLOS journals require that the minimal data set be made fully available. For more information about our data policy, please see http://journals.plos.org/plosone/s/data-availability.

Reviewers' comments:

Reviewer's Responses to Questions

**Comments to the Author**

1. Is the manuscript technically sound, and do the data support the conclusions?

Reviewer #1: Yes

Reviewer #2: Partly

2. Has the statistical analysis been performed appropriately and rigorously? 

Reviewer #1: Yes

Reviewer #2: I Don't Know

3. Have the authors made all data underlying the findings in their manuscript fully available?

Reviewer #1: Yes

Reviewer #2: Yes

4. Is the manuscript presented in an intelligible fashion and written in standard English?

Reviewer #1: Yes

Reviewer #2: No

5. Review Comments to the Author

Reviewer #1: The purpose of this manuscript is to explore the sustainable path of small and medium-sized luxury companies in Vietnam through interviews. Over all, this manuscript is thoroughly researched and the findings are complete. However, I have the following comments that I expect to Help the authors improve their manuscript.

1. In 2.1, the authors mention that "Sustainability is viewed from three perspectives: environmental, economic, and social..." I suggest that this logic can be followed in subsequent sections. In addition, I would like to see more scholarly explanations of these phenomena in addition to the current state of business. The authors need to expand the literature review in this section.

2. Based on resource dependency theory, the manuscript examines the sustainability contributions of stakeholders to luxury companies. However, the results are mostly presented as a list, whether more (hidden) information can be obtained from them is of more interest to me.

3. The topics are interesting, but the results are not very informative and attractive to the reader.

4. The contribution of the theory may be insufficient and should be better presented.

5. Smaller grammatical errors were detected, e.g., incorrect sentence structure, use of jargon, acronyms, etc. The authors are advised to proofread the draft again.

Reviewer #2: The authors focus on the practices and challenges for the sustainable fashion of luxury boutique fashion brands in Vietnam. The sustainability of luxury goods is a topic of great concern, but I still have some concerns about this article.

First, the author's research object is Luxury boutique fashion brand (LBFB). According to the author, LBFB is defined as: “Luxury boutique fashion brand (LBFB hereafter) refers to small & medium fashion businesses involved with the designing, manufacturing, and selling of stylish clothing, jewellery, or other luxury goods”. I don't understand the relationship between LBFB and luxury brands, which seems to be luxury brands with smaller scales according to the author's description? I was very confused. Then, in the part of Methodology, the authors also mention: “The selections of these luxury boutique fashion businesses are based on the matching of their product lines and prices in comparison with the large high-profile fashion brands (Louis Vuitton, Gucci, D&G) in the marketplace”. Based on this, LBFB seem to be sold at a similar price to luxury goods, and I don't see how the author can tell the difference. In general, the definition of LBFB needs more literature support and logical support.

Second, the authors did not do enough work in the literature review part. For the research questions in this paper, the author should review more relevant literature and research conclusions. the structure is sometimes unclear and there are no explanations to guide the reader through the manuscript. Instead, the reader has to figure out many things for himself. For example, the author mentions, “Similar to the high-profile luxury fashion brands, waste generation, excessive usage of natural resources, and negligence of the workforce are some of the major challenges that LBFB are facing recently (Moore, Doherty, and Doyle 2010).” However, the author then summarized totally different challenges faced by LBFB according to the interviews in the part of methods to address the research question in this paper.

In addition, the authors do not distinguish the difference between this paper and previous studies, nor does he clearly state the contributions made by this paper.

Finally, there are too many minor errors with this manuscript: misspellings, capitalization, missing/multiple words, punctuation, etc

6. PLOS authors have the option to publish the peer review history of their article (what does this mean?). If published, this will include your full peer review and any attached files.

Reviewer #1: **Yes: **Ming Yuan

Reviewer #2: No

---

## [Author Response · Author response to Decision Letter 0]

3 May 2023

Reviewer 1 Question 1. In 2.1, the authors mention that "Sustainability is viewed from three perspectives: environmental, economic, and social..." I suggest that this logic can be followed in subsequent sections. In addition, I would like to see more scholarly explanations of these phenomena in addition to the current state of business. The authors need to expand the literature review in this section.

Response to Reviewer 1 Question 1: In line with the suggestion of reviewer, we extend our literature review to include additional explanations of sustainability from three perspectives: environment, economic, and social. The revision is from page 4 to page 7 (highlighted in red) as follows:

With the increased emphasis on reducing the global warming and climate change, sustainability concepts have gained significant importance in the last two decades (Kleine & Von Hauff, 2009; Nayak et al., 2019; White et al., 2019) . The term sustainability was coined in 1987 in Brundtland report, which means satisfying the current needs without compromising the future generation’s needs (Keeble et al., 2003). Sustainability is viewed from three perspectives: environmental, economic, and social, which are known as the “Triple Bottom Line (TBL)” (Kleine & Von Hauff, 2009). Langenwalter (2009) mentioned “Sustainability involves respecting people (at all levels of the organization), and the community; respecting the planet, recognizing that resources are finite; and generating profits that arise from adhering to these principles”. 

The first aspect, environmental sustainability relates to the impacts caused due to a large amount of energy usage and water consumption; greenhouse gas (GHG) emission; hazardous waste generation; and discharge of toxic effluent containing dyes, finishes and auxiliaries to the eco-system during garment manufacturing (Turker & Altuntas, 2014). Fashion manufacturing has been recognized as one of the largest environmental polluters as several processes use large quantities of chemicals, water, and auxiliaries (Evans & Peirson-Smith, 2018). 

The second aspect, social sustainability traditionally relates to the improvement of working conditions; working hours; racism, gender equality, fair wages; health and safety risks of employees (Nayak et al., 2019). Sustainability goes beyond the relationship with the environment. It should address issues of the society, local communities, and relationships within our business operation. Thus, sustainable businesses must reduce the negative impacts on human livelihoods and well-being, with intersecting ecological, economic, and socio-political dimensions (Zeisel, 2020). The pandemic has emphasised the importance of social sustainability that ensure work-life balance for employees, local community development, and community support during and after COVID-19 to ensure healthy livings of relevant stakeholders (Euromonitor, 2022; Zeisel, 2020). However, the social aspects of sustainability have also been neglected in many countries manufacturing fashion.

The third TBL or economic aspects of fashion sustainability is related to how the business operation impact on the overall economy health of its support networks and community (Henninger et al., 2016). This aspect in fashion is mostly related to using resources in a controlled manner so that the manufacturing of fashion can be sustained indefinitely. Beard (2008) highlighted that “The major challenge in fashion sourcing lies how the souring process can be ethical and transparent, in addition to the environmental pollution, and ultimately the garment’s aftercare and disposal. 

Reviewer 1 Question 2: Based on resource dependency theory, the manuscript examines the sustainability contributions of stakeholders to luxury companies. However, the results are mostly presented as a list, whether more (hidden) information can be obtained from them is of more interest to me.

Response to Reviewer 1 Question 2: In line with the suggestion of reviewer 1, we have extended our findings including evidences and implications for resource dependence theory (page 12 to 20) with some examples as follows: 

4.1 The current practices with relevant stakeholders of sustainable fashion

Page 12: 

...Other stakeholder groups were reported namely business partners, media and government agencies. But they are considered as less important due to the limited mentioning by the respondents (less than 50%). Through collaborations and interactions with these major stakeholders, the LBFB stabilizes their sustainable business operation by exchanging constraint resources such as materials, information, know-how, localized customs. The inter-dependences allow LBFB to apply sustainable practices while also enabling other stakeholders to address their interests or concerns for their business survival or social welfares. Applying the iterative thematic coding process to the interview transcripts as to how participating LBFB integrate the sustainable factors into their various business operations with the target stakeholders (customers, local communities, local communities, business partners, etc.), three dominant themes were identified. The key themes include (i) the promotion of ethnic cultures, (ii) the enhanced usage of local community resources, and (iii) the co-creation of sustainable lifestyle....

Page 13:

 .... Ethnic culture is believed to be one of the crucial influences in fashion patterns and trends. The stakeholder of Vietnamese ethnic minorities has a strong cultural background which is reflected through their traditional elegant clothes, expressive garments, costume and other accessories. However, they have some difficulties to maintain the current practices to represent their culture as well as improvements in their standards of living. In order to manage their uncertainty, they collaborate with LBFB who has resource constraints for idea, production concepts for authentic clothes, garments, costume and other accessories. As such, LBFB preserve and promote the cultural traditions of ethnic minority groups to the public through the fashion product designs. LBFB invite artisans of constraining Vietnamese minority groups onto their business production to gain their support, to use their expertise, to get legitimacy of the authentic apparel productions. In return, artisans, representatives of minority group, ensure their cultural reflections via the design and productions of clothes, garments, costume and other accessories.

Extract 1 (SME7): “Fashion is taking out inspiration from the place that you’re in, your environment and then to showcase it, to tell a story, to send a message. Our collection is a journey in Vietnam. We want our customers to take a piece of their travel back home|

…

Page 14: 

Sustainable fashion supply chain favors the use of eco-materials in green manufacturing, distribution and retailing. In this case, globalization allows multinational corporations to locate the resources for their fashion production anywhere in the world to maximize economic of scales. Thus, they may have advantages to highlight eco-friendly products for competitive advantage. With LBFB adapting sustainability practice, the increase competitions enforce the dependence among LBFB and local suppliers to highlight the usages of local resources in sustainable fashion productions. Local resources are sourced and used in the process of manufacturing such as using talented artisans, using local silk, fabrics, and traditional handicrafts. The use of local raw materials might help to create competitive advantage for these manufacturers via creating unique fashion products. Further, this builds up the reputation for Vietnamese ethnic eco-friendly products to attract attention from potential foreign investors and visitors to the ethnic minority travel destinations. 

Extract 3 (SME4): “The limited collections made by (local) designers for instances and, then, offer the local artisans to the handmade craft. So that I can either support designers or the artisans in those villages”

…

Page 14: 

For local communities and business, most are small and micro enterprises who have limited opportunities to work with the global firms in their supply chains. Instead, forming the alliances with LBFB help them to maintain the business operations and, as a result, create stable local community living environments. This practice results a win-win situation for all involved stakeholders and, thus, contribute to the sustainable development for the brand in the local context.

…

Page 16: 

The LBFB intention to foster sustainable fashion depends on customers because they are the predominant buyer of LBFB. At the same time, customers are dependent on LBFB to provide high quality eco-friendly products for their clothes selections and fashion style practices. The dependences require both parties to co-create the sustainable practices in fashion marketplace. As such, boutique fashion businesses try to produce high quality products that can reduce the ‘wear and tear’ in order to decrease number of new purchases. Furthermore, these enterprises encourage their consumers to reuse and recycle their clothes by organizing the swap events or reward consumers with vouchers for the next purchase. This helps to raise awareness of the sustainable consumption.

Extract 6 (SME5): “We choose to convey the message of a green lifestyle to customers in various tactful and indirect ways such as clothing exchange reward programs, usage of the fabric shopping bags, and decoration of our retails stores following the eco-friendly concept”

Extract 7 (SME3): Our customers re-organize their closets to exchange them among friends and like-minded customers, or to donate for the charity during the exchange event.

Page 17:

On the hand, sustainability practice is moved to further levels where not only the use of fabric shopping bags is encouraged, but also the motivation of these practices to relevant stakeholders are also taken into consideration. While the activities aim to generate little or no environmental impact, they may generate higher costs on the businesses to date and create some concerns from the business partners and employees. The dependences of these internal stakeholders require the owners/managers of LBFB to stabilize the business operation with through socializing, discussing regarding valuable benefits for the brand, society, and environment. 

Extract 9 (SME9): “Because sustainable is a trend, to follow the trend you must have a team to really understand the sustainable concept, you need equipments, you need lots of resources, assets to be sustainable which require some supports from business partners and employees” 

Extract 10 (SME4): The production of using 0% chemicals, we need to really drive to ensure that we have got the best product to our consumers but also do no harm in the process for the workers. 

4.2 The challenges of sustainable fashion while dealing with relevant stakeholders

Page 18: 

Another aim of the research is to examine what are the challenges faced by participating LBFB in adapting sustainability practices. In analysing the interview transcripts, three dominant themes regarding the challenges were identified, namely: (i) Small and Medium Enterprise (SMEs) related challenges, (ii) consumers related challenges and (iii) macro levels related challenges. While particular resources from these stakeholders may only constitute a small part of total resource needs, they threaten the ability of LBFB to continue sustainable fashion practicing in the absence of the resources.

Reviewer 1 Question 3: The topics are interesting, but the results are not very informative and attractive to the reader.

Response to Reviewer 1 Question 3: Thanks for your feedback, we added some additional quotes and justifications from the theoretical respective of resource dependence theory from page 12 to 20 (as highlighted in red in some examples above).

Reviewer 1 Question 4: The contribution of the theory may be insufficient and should be better presented.

Response to Reviewer 1 Question 4: We thank the reviewer for the comment. We break down the original section 5. Discussions and Contributions to three sections: 

5. Discussions (page 22-25)

6. Theoretical Contributions and Practical Implications (page 25-27)

7. Limitation and Future Research (page 28)

The contributions to the theory and practices in section 6 are new and to emphasise the research gap this study has filled. The update (highlighted in red) is available in page 25-27 as follows: 

Although previous research has identified the applications of resource dependence theory, quantitative survey is the most common method used to collect data and test the theory via conceptual frameworks. There are limited empirical work that have looked at this theory in the qualitative lens to provide richer information for interdependences among stakeholders. Building on recommendations from (Hitt et al., 2016), this study made an important contribution to the literature by offering a resource dependence perspective to understand how interdependences works with SMEs in Vietnam fashion market. Due to the small scale of business, LBFB has some limitations for business developments in order to compete the global fashion brands. Further, their sustainable development requires lots of resources (financial, technical, and manpower, for example) that they may lack. Thus, they will be very likely to undertake new ways to solve problems via networking and collaboration. Resource Dependences Theory argue that the fewer the number of resources, the more numerous the connections and interdependencies among stakeholders arise. Further, there are two angles of interdependences: 1) filing the resource gaps of the business, and 2) improving the usages of existing resources. In this study, collaborations with relevant stakeholders will help the brand to fill the gaps of resource constraints. New knowledge, expertise and resources from local community businesses, artisans, employees enable LBFB to develop their eco-friendly, authentic fashion products. Moreover, LBFB work well with their customers to co-create the sustainable lifestyle via good product quality, clothes exchange programs, and eco-friendly store concept integration. The interdependencies will help LBFB overcome the disadvantages of smallness and limited resources to better address the needs of customers. On the other hand, working with LBFB allow relevant stakeholders to overcome obstacles associated with their limited network and market knowledge. It helps local businesses, Vietnamese minority groups to improve the usage of their existing resources (labour forces, authentic designs, local silk, fabrics, and traditional handicrafts, etc.). 

Regarding the challenges, the dynamic nature of external market environment requires LBFB to seek more supports from other stakeholders. Thus, holding some valuable resources was a necessary but insufficient condition for the business to achieve competitive advantages in the marketplace (Hitt et al., 2016). To address these issues, LBFB need additional efforts to educate customers for the practice, to seek helps from the governments for the certificate achievements and supporting policies to convert resource portfolio into their capabilities in the marketplace. In the meantime, they still struggle to maintain their competitive advantages in the marketplace while doing good enough for the sustainable development. These results highlight the implication of Resource Dependences Theory in the context of sustainable fashion SMEs. 

There are few practical implications from the study. First, the development of sustainable fashions are challenging for LBFB due to intense competitions by some powerful global brands. LBFB businesses are often unable to obtain sufficient resources themselves for sustainable business operation. Instead, LBFB should possess resources from relevant stakeholders including local businesses, artisans from cultural richness minority groups (manpower, authentic designs and fabrics with traditional handicrafts). LBFB can benefit from these stakeholder participations in the supply chain for the product design and quality improvements. LBFB should take advantage of business partner knowledge to improve their business operation and capabilities and better align products with local market trends. 

Second, the improvement of product quality and ‘local touch’ customer services could increase customer engagement with the brand to co-create the sustainable lifestyles. Durable product quality not only help customers reduce their frequent purchase of fashion items, but it also encourages them to join clothes exchange programs organized by LBFB, to share the sustainable concept with other like-minded people and, in turn, can promote the brand implicitly. It allows customers to reduce their fast fashion purchase habits and be more sustainable for fashion product consumptions. Third, while there are some good sights for the sustainable fashion outlooks for LBFB in Vietnam, they should know the existing challenges such as constraint eco-materials in the local community, sustainable fashion practice readiness among mass customers, and limited supports from the authorities. If LBFB want to scale up their business, they can’t simply rely on local resources for productions and operation. Many local businesses are not ready with the larger volume for productions with consistent good quality. Moreover, they should spend more efforts to educate customers for sustainable fashion concept as well as spend more resources for some certificates. Thus, maintaining the current scales of business with niches customer segments maybe a good option for LBFB to grow slowly but sustainably in this market condition. 

Reviewer 1 Question 5: Smaller grammatical errors were detected, e.g., incorrect sentence structure, use of jargon, acronyms, etc. The authors are advised to proofread the draft again.

Response to Reviewer 1 Question 5: We thank the reviewer for these comments. We did proofread the revised manuscript to fix all grammatical errors. 

Reviewer 2 Question 1: First, the author's research object is Luxury boutique fashion brand (LBFB). According to the author, LBFB is defined as: “Luxury boutique fashion brand (LBFB hereafter) refers to small & medium fashion businesses involved with the designing, manufacturing, and selling of stylish clothing, jewelry, or other luxury goods”. I don't understand the relationship between LBFB and luxury brands, which seems to be luxury brands with smaller scales according to the author's description? I was very confused. 

Then, in the part of Methodology, the authors also mention: “The selections of these luxury boutique fashion businesses are based on the matching of their product lines and prices in comparison with the large high-profile fashion brands (Louis Vuitton, Gucci, D&G) in the marketplace”. Based on this, LBFB seem to be sold at a similar price to luxury goods, and I don't see how the author can tell the difference. In general, the definition of LBFB needs more literature support and logical support.

Response to Reviewer 2 Question 1: We thank the reviewer for these comments. LBFB is more about the small and medium enterprise (SMEs) in fashion business who offer luxury fashion products. They are small in scale but they have some unique authentic designs. The brands would be in the niche and high-fashion categories, with clothing and handbags being the main products. Some examples were Vo Viet Chung (Vietnam), Afro Street (UK), dotcomme (Australia) to name a few. We added some additional explanations in the introduction section (page 2-3 – highlighted in red) as follows:

Luxury boutique fashion brand (LBFB hereafter) refers to small & medium fashion enterprises (SMEs) involved with the designing, manufacturing, and selling of stylish clothing, jewellery, or other luxury goods. LBFBs represent a relatively small sector compared to the global luxury fashion brands and even fast fashion brands. Unlike the global luxury brands, LBFBs are limited in product styles, operate in smaller spaces, and deal with smaller production volumes (Kerr-Crowley 2022). Therefore, the product cost may be higher, which necessitates careful sourcing of inventory to manufacture the luxury product (Baker 2022). Some LBFBs manufacture their own products with authentic in-house designs with luxury materials (Thuy 2023). 

Reviewer 2 Question 2:

Second, the authors did not do enough work in the literature review part. For the research questions in this paper, the author should review more relevant literature and research conclusions. the structure is sometimes unclear and there are no explanations to guide the reader through the manuscript. 

Response to Reviewer 2 Question 2:

In line with the suggestion of reviewer, we extend our literature review to include additional explanations of sustainability from three perspectives: environment, economic, and social. The revision is from page 4 to page 7 (highlighted in red) as follows:

With the increased emphasis on reducing the global warming and climate change, sustainability concepts have gained significant importance in the last two decades (Kleine & Von Hauff, 2009; Nayak et al., 2019; White et al., 2019) . The term sustainability was coined in 1987 in Brundtland report, which means satisfying the current needs without compromising the future generation’s needs (Keeble et al., 2003). Sustainability is viewed from three perspectives: environmental, economic, and social, which are known as the “Triple Bottom Line (TBL)” (Kleine & Von Hauff, 2009). Langenwalter (2009) mentioned “Sustainability involves respecting people (at all levels of the organization), and the community; respecting the planet, recognizing that resources are finite; and generating profits that arise from adhering to these principles”. 

The first aspect, environmental sustainability relates to the impacts caused due to a large amount of energy usage and water consumption; greenhouse gas (GHG) emission; hazardous waste generation; and discharge of toxic effluent containing dyes, finishes and auxiliaries to the eco-system during garment manufacturing (Turker & Altuntas, 2014). Fashion manufacturing has been recognized as one of the largest environmental polluters as several processes use large quantities of chemicals, water, and auxiliaries (Evans & Peirson-Smith, 2018). 

The second aspect, social sustainability traditionally relates to the improvement of working conditions; working hours; racism, gender equality, fair wages; health and safety risks of employees (Nayak et al., 2019). Sustainability goes beyond the relationship with the environment. It should address issues of the society, local communities, and relationships within our business operation. Thus, sustainable businesses must reduce the negative impacts on human livelihoods and well-being, with intersecting ecological, economic, and socio-political dimensions (Zeisel, 2020). The pandemic has emphasised the importance of social sustainability that ensure work-life balance for employees, local community development, and community support during and after COVID-19 to ensure healthy livings of relevant stakeholders (Euromonitor, 2022; Zeisel, 2020). However, the social aspects of sustainability have also been neglected in many countries manufacturing fashion.

The third TBL or economic aspects of fashion sustainability is related to how the business operation impact on the overall economy health of its support networks and community (Henninger et al., 2016). This aspect in fashion is mostly related to using resources in a controlled manner so that the manufacturing of fashion can be sustained indefinitely. Beard (2008) highlighted that “The major challenge in fashion sourcing lies how the souring process can be ethical and transparent, in addition to the environmental pollution, and ultimately the garment’s aftercare and disposal.

Reviewer 2 Question 3:

Instead, the reader has to figure out many things for himself. For example, the author mentions, “Similar to the high-profile luxury fashion brands, waste generation, excessive usage of natural resources, and negligence of the workforce are some of the major challenges that LBFB are facing recently (Moore, Doherty, and Doyle 2010).” However, the author then summarized totally different challenges faced by LBFB according to the interviews in the part of methods to address the research question in this paper.

Response to Reviewer 2 Question 3: 

Thanks for the reviewer to point out this issue. Actually, the claims were made for years and fashion brands (both global and LBFB) still working on the issues. To make it clear, we revise the sentences as follows:

Similar to the high-profile luxury fashion brands, waste generation, excessive usage of natural resources, and negligence of the workforce are some of the major challenges that LBFB needed to deal with (Moore, Doherty, and Doyle 2010) 

Reviewer 2 Question 4: 

In addition, the authors do not distinguish the difference between this paper and previous studies, nor does he clearly state the contributions made by this paper.

Response to Reviewer 2 Question 4: 

We thank the reviewer for the comment. We break down the original section 5. Discussions and Contributions to three sections: 

5. Discussions (page 22-25)

6. Theoretical Contributions and Practical Implications (page 25-27)

7. Limitation and Future Research (page 28)

The contributions to the theory and practices in section 6 are new and to emphasise the research gap this study has filled. The update (highlighted in red) is available in page 25-27 as follows: 

Although previous research has identified the applications of resource dependence theory, quantitative survey is the most common method used to collect data and test the theory via conceptual frameworks. There are limited empirical work that have looked at this theory in the qualitative lens to provide richer information for interdependences among stakeholders. Building on recommendations from (Hitt et al., 2016), this study made an important contribution to the literature by offering a resource dependence perspective to understand how interdependences works with SMEs in Vietnam fashion market. Due to the small scale of business, LBFB has some limitations for business developments in order to compete the global fashion brands. Further, their sustainable development requires lots of resources (financial, technical, and manpower, for example) that they may lack. Thus, they will be very likely to undertake new ways to solve problems via networking and collaboration. Resource Dependences Theory argue that the fewer the number of resources, the more numerous the connections and interdependencies among stakeholders arise. Further, there are two angles of interdependences: 1) filing the resource gaps of the business, and 2) improving the usages of existing resources. In this study, collaborations with relevant stakeholders will help the brand to fill the gaps of resource constraints. New knowledge, expertise and resources from local community businesses, artisans, employees enable LBFB to develop their eco-friendly, authentic fashion products. Moreover, LBFB work well with their customers to co-create the sustainable lifestyle via good product quality, clothes exchange programs, and eco-friendly store concept integration. The interdependencies will help LBFB overcome the disadvantages of smallness and limited resources to better address the needs of customers. On the other hand, working with LBFB allow relevant stakeholders to overcome obstacles associated with their limited network and market knowledge. It helps local businesses, Vietnamese minority groups to improve the usage of their existing resources (labour forces, authentic designs, local silk, fabrics, and traditional handicrafts, etc.). 

Regarding the challenges, the dynamic nature of external market environment requires LBFB to seek more supports from other stakeholders. Thus, holding some valuable resources was a necessary but insufficient condition for the business to achieve competitive advantages in the marketplace (Hitt et al., 2016). To address these issues, LBFB need additional efforts to educate customers for the practice, to seek helps from the governments for the certificate achievements and supporting policies to convert resource portfolio into their capabilities in the marketplace. In the meantime, they still struggle to maintain their competitive advantages in the marketplace while doing good enough for the sustainable development. These results highlight the implication of Resource Dependences Theory in the context of sustainable fashion SMEs. 

There are few practical implications from the study. First, the development of sustainable fashions are challenging for LBFB due to intense competitions by some powerful global brands. LBFB businesses are often unable to obtain sufficient resources themselves for sustainable business operation. Instead, LBFB should possess resources from relevant stakeholders including local businesses, artisans from cultural richness minority groups (manpower, authentic designs and fabrics with traditional handicrafts). LBFB can benefit from these stakeholder participations in the supply chain for the product design and quality improvements. LBFB should take advantage of business partner knowledge to improve their business operation and capabilities and better align products with local market trends. 

Second, the improvement of product quality and ‘local touch’ customer services could increase customer engagement with the brand to co-create the sustainable lifestyles. Durable product quality not only help customers reduce their frequent purchase of fashion items, but it also encourages them to join clothes exchange programs organized by LBFB, to share the sustainable concept with other like-minded people and, in turn, can promote the brand implicitly. It allows customers to reduce their fast fashion purchase habits and be more sustainable for fashion product consumptions. Third, while there are some good sights for the sustainable fashion outlooks for LBFB in Vietnam, they should know the existing challenges such as constraint eco-materials in the local community, sustainable fashion practice readiness among mass customers, and limited supports from the authorities. If LBFB want to scale up their business, they can’t simply rely on local resources for productions and operation. Many local businesses are not ready with the larger volume for productions with consistent good quality. Moreover, they should spend more efforts to educate customers for sustainable fashion concept as well as spend more resources for some certificates. Thus, maintaining the current scales of business with niches customer segments maybe a good option for LBFB to grow slowly but sustainably in this market condition. 

Reviewer 2 Question 5: 

Finally, there are too many minor errors with this manuscript: misspellings, capitalization, missing/multiple words, punctuation, etc. 

Response to Reviewer 2 Question 5: We thank the reviewer for these comments. We did proofread the revised manuscript to fix all grammatical errors.

---

## [Decision Letter · Decision Letter 1]

6 Jun 2023

The Practice of Sustainable Fashion of Luxury Boutique Fashion Brands in Vietnam: What go right, and what go wrong

PONE-D-23-01713R1

Dear Dr. Nguyen,

We’re pleased to inform you that your manuscript has been judged scientifically suitable for publication and will be formally accepted for publication once it meets all outstanding technical requirements.

Kind regards,

Han Lin

Academic Editor

PLOS ONE

Additional Editor Comments (optional):

Reviewers' comments:

Reviewer's Responses to Questions

**Comments to the Author**

1. If the authors have adequately addressed your comments raised in a previous round of review and you feel that this manuscript is now acceptable for publication, you may indicate that here to bypass the “Comments to the Author” section, enter your conflict of interest statement in the “Confidential to Editor” section, and submit your "Accept" recommendation.

Reviewer #1: All comments have been addressed

Reviewer #2: All comments have been addressed

2. Is the manuscript technically sound, and do the data support the conclusions?

Reviewer #1: Yes

Reviewer #2: Yes

3. Has the statistical analysis been performed appropriately and rigorously? 

Reviewer #1: Yes

Reviewer #2: Yes

4. Have the authors made all data underlying the findings in their manuscript fully available?

Reviewer #1: Yes

Reviewer #2: Yes

5. Is the manuscript presented in an intelligible fashion and written in standard English?

Reviewer #1: Yes

Reviewer #2: Yes

6. Review Comments to the Author

Reviewer #1: (No Response)

Reviewer #2: The manuscript reads much better now. I commend the authors for taking on board the suggestion and significantly improving the manuscript. Overall, the manuscript now makes a structured and clear impression in terms of content.

7. PLOS authors have the option to publish the peer review history of their article (what does this mean?). If published, this will include your full peer review and any attached files.

Reviewer #1: No

Reviewer #2: No

---

## [Editor Report · Acceptance letter]

12 Jun 2023

PONE-D-23-01713R1 

The practice of sustainable fashion of luxury boutique fashion brands in Vietnam: what go right, and what go wrong 

Dear Dr. Nguyen:

I'm pleased to inform you that your manuscript has been deemed suitable for publication in PLOS ONE. Congratulations! Your manuscript is now with our production department. 

Kind regards, 

on behalf of

Dr. Han Lin 

Academic Editor

PLOS ONE